# Randomized Feature Squeezing against Unseen Attacks without Adversarial Training

## Abstract

Deep learning has made tremendous progress in the last decades; however, it is not robust to adversarial attacks. Perhaps the most effective approach for this is adversarial training, although it is impractical as it needs prior knowledge about the attackers and incurs high computational costs. In this paper, we propose a novel approach that can train a robust network only through standard training with clean images without awareness of the attacker's strategy. We add a specially designed network input layer, which accomplishes a randomized feature squeezing to reduce the malicious perturbation. It achieves the state of the art of robustness against unseen $l_1, l_2$ and $l_\infty$ attacks at one time in terms of the computational cost of the attacker versus the defender through just 100/50 epochs of standard training with clean images in CIFAR-10/ImageNet. Both experiments and Rademacher complexity analysis validate the high performance. Moreover, it can also defend against the "attacks" on training data, i.e., unlearnable examples, seemingly being the only solution for the One-Pixel Shortcut without any data augmentation.

## 1 Introduction

The vulnerability of neural networks has been widely acknowledged by the deep learning community since the seminal work of Szegedy et al. (2014). A lot of solutions have been proposed to solve these problems. They can be categorized into three classes.

The first is preprocessing-based approaches which include bit-depth reduction (Xu et al., 2018), JPEG compression, total variance minimization, image quilting (Guo et al., 2018), and Defense-GAN (Samangouei et al., 2018). With this kind of preprocessing, the hope is that the adversarial effect can be reduced. However, it neglects the fact that the adversary can still take this operation into account and craft an effective attack through Backward Pass Differentiable Approximation (BPDA) (Athalye et al., 2018).

Secondly, perhaps the most effective method is adversarial training. In the training phase, the attack is mimicked through the backward gradient propagation with respect to the current network state. There is a large volume of work that falls into this class which differs in how to generate extra training samples. Madry et al. (2018) used a classical 7-step PGD attack, while other approaches are also possible, such as Mixup inference (Pang et al., 2020), feature scattering (Zhang & Wang, 2019), feature denoising (Xie et al., 2019), geometry-aware instance reweighting (Zhang et al., 2021), and channel-wise activation suppressing (Bai et al., 2021). External (Gowal et al., 2020) or generated data (Gowal et al., 2021; Rebuffi et al., 2021) are also beneficial for robustness. The inherent drawbacks are the large computation cost and the need for prior knowledge about attacks. This is certainly not realistic in practice. Also, there is a possibility of robust overfitting (Rice et al., 2020).

The last is adaptive test-time defenses. They try to purify the input in an iterative way as in Mao et al. (2021); Shi et al. (2021); Yoon et al. (2021) or adapt the model parameters or even network structures to reverse the attack effect. For example, close-loop control is adopted in Chen et al. (2021), and a neural Ordinary Differential Equation (ODE) layer is applied in Kang et al. (2021). Unfortunately, most of them are proven to be not effective in Croce et al. (2022).

It turns out the progress is not optimistic, and even 1%-2% improvement on AutoAttack(Croce & Hein, 2020a) requires huge computational cost and moreover, not effective for unseen attacks. Here we ask a question: "can we design a novel network and loss function thereof that can drive the

network to be robust on its own without awareness of adversarial attacks?" In other words, we do not intend to generate extra adversarial samples like most other approaches do, and standard training with clean images is enough. Indeed, there should be no prior knowledge of attacks needed at all. This certainly poses a great challenge to the construction of networks as it is not clear even whether it is feasible. On the other hand, it appears to be possible since deep networks have a very high capacity. Unfortunately, Ilyas et al. (2019) pointed out network tends to learn discriminant features that can help correct classification, regardless of robustness. It motivates us to take the point of view from the network input side. How can we make a new input layer that is most suitable for network robustness? Our intuition is essentially very simple. As attacks can always walk across the class decision boundary through the malicious feature perturbations, feature squeezing might be helpful, at least reducing the space of being altered, which, indeed, is supported by Rademacher complexity Yin et al. (2019) in adversarial setting which depends on input feature dimension. However, fundamentally different from the preprocessing work Xu et al. (2018), the input features are randomized squeezed with parameters learned during training as shown in Figure 1. Moreover, in test we simplify this layer and greatly facilitate the evaluation. The experiments of CIFAR-10 and ImageNet demonstrate this approach is very useful in promoting robustness of networks. It needs to be pointed out that although our main motivation is adversarial defense against unseen attacks, it turns out that ours is much less influenced by the unlearnable examples, i.e., data intentionally manipulated for unauthorized usage for training DNNs. Recently, One-Pixel Shortcut(OPS) has been proposed in Wu et al. (2023) and could effectively degrade model accuracy even to almost an untrained counterpart even equipped with adversarial training, while ours sustain around 60%.

To our knowledge, we are the first to clear a tough bar by enhance robustness on unseen attacks without adversarial training. With all the source codes and pre-trained models online A.1, our work has the following contributions:

- We design a special input layer that uses the reciprocal and multiplication to implement our randomized feature squeezing, which has not been investigated before. Furthermore, it could be easily plunged into different networks such as WideResNet and ConvNeXt to boost their performance.

- **Our work is the only one that does not require any prior knowledge about the attacks with standard training with clean images;** while significantly outperforms adversarial ones with training costs up to 1-5 orders of magnitude higher than ours, for black-box and $l_1, l_2$-attacks in both CIFAR-10 and ImageNet.

- **By transforming our random network into a deterministic one, we draw a connection between our robust accuracy and Rademacher complexity, which guarantees our defense is fully legitimate.**

- Our approach appears to be the only one that can effectively deal with unlearnable examples generated by the state of the art OPS without any data augmentation.

## 2 RELATED WORKS

There are some works that add some extra preprocessing steps. For example, in Yang et al. (2019), pixels are randomly dropped and then reconstructed using matrix estimation. Ours is not preprocessing. We just add an extra layer inside the network, and the network is trained and tested as usual without explicit image completion. Besides this, to get high robust accuracy, Yang et al. (2019) needs adversarial training while we adopt standard training with clean images.

Another related work is certified adversarial robustness via randomized smoothing (Cohen et al., 2019). The base classifier is trained with Gaussian data augmentation, and inference is based on the most likely class of the input perturbed by isotropic Gaussian noise. Ours is based on standard training and testing, and there is no perturbation-based training data augmentation involved at all.

Stochastic Neural Networks(SNNs) (Eustratiadis et al., 2021; Däubener & Fischer, 2022; Lee et al., 2023) achieve robustness by intentionally injecting noise to the intermediate layers of the preexisting networks, which is very different from ours. We are motivated by the inherent weakness of the current network design and are trying to modify it such that adversarial defense can be implemented. Moreover, it seems that SNNs are far behind the top-ranking methods in the robustbench leaderboard at https://robustbench.github.io/, which we compared with. Moreover, there is no SNNs for ImageNet.

Recently, there are some works that address the robustness from the network architecture's perspective. Wu et al. (2021) investigates impact of the network width on the model robustness, and proposes Width Adjusted Regularization. Similarly, Huang et al. (2021) explores architectural ingredients of adversarially robust deep neural networks in a thorough manner. Liu et al. (2023a) established that the higher weight sparsity is beneficial for adversarially robust generalization via Rademacher complexity. Wang et al. (2022) proposes batch normalization removal, such that adversarial training can be improved. Singla et al. (2021) shows that using activation functions with low curvature values reduces both the standard and robust generalization gaps in adversarial training. They are in some sense similar to ours, but our motivations are fundamentally different. There is no adversarial training involved in our approach at all.

There are some attempts (Tramèr & Boneh, 2019; Maini et al., 2020; Croce & Hein, 2022; Laidlaw et al., 2021; Dai et al., 2022b) to deal with multiple attacks simultaneously. **Among them, the only relevant works for unseen attacks are Laidlaw et al. (2021); Dai et al. (2022b), however they adopt costly adversarial training, and only for CIFAR-10.** There are also some benchmarks (Dai et al., 2023; Kang et al., 2019) which extend beyond the $l_p$ attacks.

Adversarial purification is another research line to defend against unseen attacks, but it is extremely slow. For example, in Table 14 of Nie et al. (2022), the inference time is around (100-300)x of standard one. Also, they need pre-trained diffusion models, which are very expensive to train. Another big disadvantage is that the evaluation of robustness for these methods is very difficult due to very high memory consumption. Moreover, adversarial purification cannot deal with OPS, as the underlying static model has very low clean accuracy.

## 3 BACKGROUND

A standard classification can be described as follows:

$$\min_{\vartheta} E_{(x,y) \sim D} \left[ L\left( x, y, \vartheta \right) \right], \tag{1}$$

where data examples $x \in R^d$ and corresponding labels $y \in [k]$ are taken from the underlying distribution $D$, and $\vartheta \in R^p$ is the model parameters to be optimized with respect to an appropriate function $L$, for instance cross-entropy loss. When $x \in R^d$ can be maliciously manipulated within a set of allowed perturbations $S \subseteq R^d$, which is usually chosen as a $l_p$-ball ($p \in \{1, 2, \infty\}$) of radius $\epsilon$ around $x$, Equation 1 should be modified as:

$$\min_{\vartheta} E_{(x,y) \sim D} \left[ \max_{\delta \in S} L\left( x + \delta, y, \vartheta \right) \right]. \tag{2}$$

An adversary implements the inner maximization via various white-box or black-box attack algorithms, for example, APGD-ce (Croce & Hein, 2020a) or Square Attack (Andriushchenko et al., 2020) . The basic multi-step projected gradient descent (PGD) is

$$x^{t+1} = \Pi_{x+S} \left( x^t + \alpha \mathrm{sgn} \left( \nabla_x L\left( x, y, \vartheta \right) \right) \right), \tag{3}$$

where $\alpha$ denotes a step size and $\Pi$ is a projection operator. In essence, it uses the current gradient to update $x^t$, such that a better adversarial sample $x^{t+1}$ can be obtained. Some heuristics can be used to get better gradient estimation in Croce & Hein (2020a). On the other hand, outer minimization is the goal of a defender.

Adversarial training is the most effective approach to achieve this outer minimization via augmenting the training data with crafted samples. In fact, all current approaches, including test-time adaptive defense as it needs a base classifier, aim to learn the parameters of a pre-existing model to improve the robustness. In this paper, we try to increase the robustness through a specially designed input layer such that standard training with clean images can be adopted.

## 4 METHOD

### 4.1 INPUT LAYER

As we stated earlier, the goal of input layer is to squeeze the input feature in a random and controlled way. The whole procedure is depicted in Figure 1.

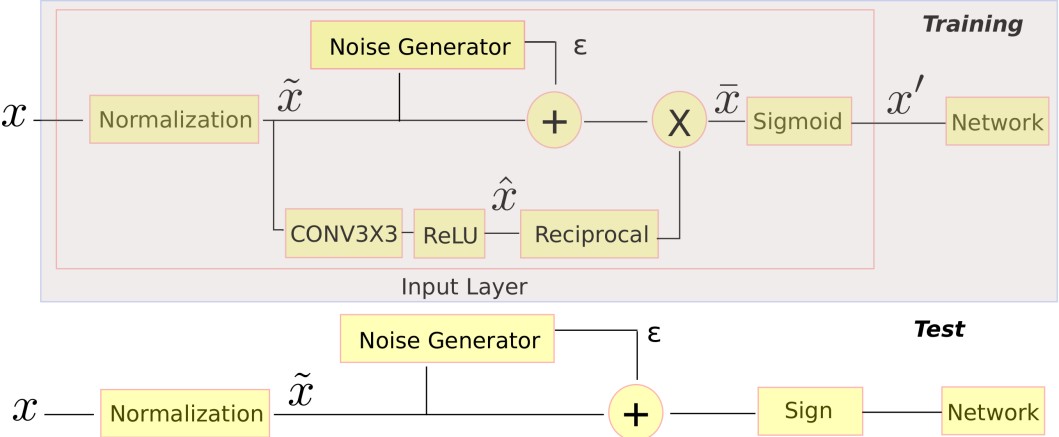

Figure 1: The shaded and non-shaded areas show the training and test framework respectively. In training, our specially designed input layer is inside the red rectangle. The input image $x$ is first normalized, then undergoes two paths. On one path, signal dependent Gaussian noise $\epsilon$ is added, and the other path includes $3 \times 3$ convolution and ReLU followed by reciprocal. Finally, these two terms are combined through multiplication and the result feeds to the Sigmoid. The final $x'$ will be used as inputs to the classification network, the same as other training approaches. End-to-end training scheme is adopted to learn the parameters of $3 \times 3$ convolution. In test, the CONV3$\times$3 path is removed wholly, and Sigmoid is replaced with Sign defined in Equation 6.

It consists of the following steps:

1. The input $x$ with $r, g, b$ channel will be normalized to a variable with a mean 0 and a standard deviation 1, through $\tilde{x} = \frac{x-mean}{std}$ in the input layer, where $mean$ and $std$ are mean and standard deviation of training set. Then, it goes through top and bottom paths.

2. In the top path, each element of $\tilde{x}$ is corrupted independently by additive Gaussian noise $\varepsilon$, where $\varepsilon \sim N\left(\tilde{x}, (\sigma(\tilde{x}))^2 I\right)$, where $\sigma$ depends on $\tilde{x}$ .

3. In the bottom path, $\tilde{x}$ goes through a $3 \times 3$ 2D convolution and ReLU, and we get $\hat{x}$ with three channels, and then its reciprocal $\frac{1}{\hat{x}+\gamma}$, where $\gamma$ is a small constant in order to make the denominator always positive, which is $1\times10^{-5}$ in this paper.

4. The top and bottom paths are combined by multiplication, i.e., $\bar{x} = (\tilde{x} + \varepsilon) \times \frac{1}{\hat{x}+\gamma}$.

5. The final output $x'$ is a Sigmoid of the $\bar{x}$, i.e., $x' = \frac{1}{1+\exp(-\bar{x})}$.

So essentially,

$$x' = \frac{1}{1 + \exp\left(-\frac{(\tilde{x}+\varepsilon)}{\hat{x}+\gamma}\right)}. \tag{4}$$

This formula can be interpreted this way. $\tilde{x} + \varepsilon$ is a polluted version of the input image, and $\frac{1}{\hat{x}+\gamma}$ tries to modulate the image based on the $\hat{x}$, named as sampling matrix having the same size as input $x$.

The key motivation is that if we enforce $\hat{x}$ to be very small through some loss function, $\left|\frac{(\tilde{x}+\varepsilon)}{\hat{x}+\gamma}\right|$ will become big and the response of Sigmoid will be on the saturated region, i.e., most elements of $x'$ will be either 0 or 1. In other words, the input feature will be squeezed in a random manner where the parameters of sampling matrix $\hat{x}$ are learned on the end-to-end training.

Accordingly, the role that $\varepsilon$ plays is just to mimic the attack that adversary may launch. Since the denominator of $\frac{(\tilde{x}+\varepsilon)}{\hat{x}+\gamma}$ is very small, usually in the order of $1\times10^{-4}$ and can't be influenced in any useful way, the only effective way is to change the sign of numerator which ends up with reverse of 0

and 1 of $x'$. Of course, if $|\tilde{x}|$ is small, little $\sigma$ is enough to switch the sign $\tilde{x}$. To that end, $\sigma(\tilde{x})$ is defined as:

$$\sigma(\tilde{x}) = \begin{cases} \kappa \times |\tilde{x}|, \text{if } \kappa \times |\tilde{x}| < \tau, \\ \tau, \quad \text{otherwise.} \end{cases} \tag{5}$$

Here, $\kappa$ is some scalar and $\tau$ is the maximum of allowed $\sigma$, as the big $\sigma$ will degrade the clean accuracy.

Based on our analysis above, one may raise a big concern regarding the obfuscated gradients Athalye et al. (2018) which may be incurred by reciprocal and Sigmoid operator in robustness evaluation. On one hand, $\hat{x}$ is very small, so the gradient of reciprocal $\frac{1}{\hat{x}+\gamma}$ will be very big. On the other hand, $\bar{x} = (\tilde{x} + \varepsilon) \times \frac{1}{\hat{x}+\gamma}$ will reside on the saturated domain of Sigmoid, i.e., the gradients of $x'$ with respect to $\bar{x}$ will be very small. Actually, this might also cause some trouble in training, as we need to learn the parameters of CONV3×3 for sampling matrix $\hat{x}$, although they might be canceled out by each other to some extent, as they are on the same path in backward pass gradient propagation.

To resolve this, in training we adopt the BPDA-like optimization procedure. Namely, for the forward pass, we evaluate the reciprocal and Sigmoid as usual, however, in the backward pass, the gradient of the reciprocal is set to be -1, and 1 for Sigmoid. While in test, because Sigmoid often goes to two extreme values 0 and 1, dependent of the sign of $\tilde{x} + \varepsilon$, we just remove the bottom path wholly, and replace the Sigmoid with Sign which is defined as:

$$Sign(x) = \begin{cases} 0, & \text{if } x < 0, \\ 0.5, & \text{if } x = 0, \\ 1, & \text{if } x > 0. \end{cases} \tag{6}$$

This will greatly simplify our robust evaluation. Since Sign is non-differentiable, the derivative of smooth $Sigmoid(ax)$ is chosen as backward gradients approximation, where $a$ is set to obtain worst robust accuracy to make sure our evaluation is legitimate. In this paper, $a = 5$.

### 4.2 LOSS FUNCTION

As mentioned earlier, we have to design a loss function to implement our motivation to make the sampling matrix $\hat{x}$ small. For each $\hat{x}$, we get $S$, the average of all the elements of $\hat{x}$ that are greater than some threshold $\beta$. Formally,

$$S = \frac{\sum_{i \in T} \hat{x}_i}{\#T}, \ \ where \ T = \{i|\ \hat{x}_i > \beta\}. \tag{7}$$

A small $\beta$ means $\hat{x}$ will become sparse. The final loss function is:

$$L = \alpha \times L_{ce} + S, \tag{8}$$

where $L_{ce}$ is cross-entropy loss, and $\alpha$ is the weight. When $\alpha$ becomes large, the loss function falls back to standard cross-entropy. In summary, there are only four parameters, $\kappa$ and $\tau$ in Equation 5, threshold $\beta$, and weight $\alpha = 0.1$ in this paper.

### 4.3 LAST MOVE

We emphasize here that since our approach is random, a same sample could be classified with different logits when executed multiple times. It will seriously mislead attackers, which will report a wrong robust accuracy. For that reason, we always take the last-move advantage. In other words, in test time, we always take the adversarial samples generated by attackers and feed them to our network once again to test. We think that is fair in practice. Attackers can always take an arbitrarily long time to figure out a malicious sample, but they only have one chance to submit it. It is the attacker's responsibility to provide a stable adversarial sample, and the last move is always the defender's privilege. In fact, other works, for example Wang et al. (2021), also adopt this under different contexts. Some theoretical analysis is provided in Däubener & Fischer (2022) for this setting. Please refer to Appendix A.2 for more explanations.

Table 1: AutoAttack comparison on CIFAR-10 (WideResNet-28-10 only except ResNet-50 in Laidlaw et al. (2021)). * denote models that are trained with $l_2$-$\epsilon$=0.5, while # with $l_\infty$-$\epsilon$=8/255; both * and # need extra training data. $l_\infty$-$\epsilon$=8/255, 16/255; $l_1$-$\epsilon$=12 and $l_2$-$\epsilon$=2. The bold indicates the best for each column. For clean and AA, we report mean and standard deviation for five runs. **Reemphasize that there is no work for unseen attacks with standard training, and if adversarial training is allowed, Laidlaw et al. (2021) and Dai et al. (2022b) (in bold) are the only two to this end.**

| Paper | Clean | AA-$l_\infty$ | | AA-$l_1$ | AA-$l_2$ | Square-$l_\infty$ | | Square-$l_1, l_2$ | |
|---|---|---|---|---|---|---|---|---|---|
| Wang et al. (2023)# | 92.44 | **67.31** | 25.46 | 10.23 | 1.18 | 73.57 | 40.28 | 35.77 | 30.78 |
| Gowal et al. (2021)# | 87.50 | 63.38 | 27.91 | 10.85 | 1.94 | 68.90 | 40.91 | 35.71 | 30.15 |
| Dai et al. (2022a)# | 87.02 | 61.55 | 26.28 | 11.22 | 1.98 | 66.99 | 38.86 | 37.15 | 30.26 |
| Wang et al. (2023)* | 95.16 | 49.33 | 3.86 | 46.08 | 6.59 | 67.02 | 18.69 | 69.38 | 44.20 |
| Rebuffi et al. (2021)* | 91.79 | 47.83 | 5.04 | 42.80 | 8.23 | 62.45 | 19.73 | 65.66 | 42.66 |
| **Laidlaw et al. (2021)** | 82.40 | 30.20 | 4.50 | 32.40 | 7.10 | 46.40 | 15.30 | 53.30 | 34.20 |
| **Dai et al. (2022b)** | 72.73 | 49.94 | 20.23 | 6.36 | 0.80 | 54.18 | 31.65 | 30.62 | 27.25 |
| Ours(mean) | 82.01 | 66.98 | **32.27** | **69.02** | **16.91** | **81.63** | **79.12** | **82.28** | **81.37** |
| Ours(std) | 0.27 | 0.29 | 0.16 | 0.22 | 0.32 | | | | |
| Ours($\sigma = 0$) | 83.82 | 6.30 | 0.54 | 0.20 | 0.00 | | | | |

# 5 EXPERIMENTS

To verify the effectiveness of our approach, we conducted the experiments on CIFAR-10 and ImageNet. We evaluate on AutoAttack and Square Attack of $l_\infty$, $l_1$ and $l_2$. AutoAttack is comprised of four attacks, namely Auto-PGD for cross-entropy and Difference of Logits Ratio (DLR) loss, FAB-attack (Croce & Hein, 2020b) and the black-box Square Attack (Andriushchenko et al., 2020), and commonly used as a robustness evaluator. Square is used as a representative black-box attack separately as well, as it is of practical significance.

## 5.1 CIFAR-10

### 5.1.1 ADVERSARIAL ROBUSTNESS

In this paper, we choose the wide residual network WideResNet-28-10 (Zagoruyko & Komodakis, 2016) as the base network, where we add our specially designed input layer as described in Section 4 with $\kappa = 2$, $\tau = 0.5$ and $\beta = 0.2$. The initial learning rate of 0.1 is scheduled to drop at 30, 60, and 80 out of 100 epochs in total with a decay factor of 0.2. The weight decay factor is set to $5 \times 10^{-4}$, and the batch size is 200. To emphasize again, we only perform standard training through just 100 epochs.

In Table 1, we compare our method with some state of the arts, which are all based on adversarial training. $l_\infty$, $l_1$ and $l_2$-AutoAttack (Croce & Hein, 2020a) are adopted. All these models are trained with one particular type of attack either with $l_\infty$-$\epsilon = 8/255$ or $l_2$-$\epsilon = 0.5$, except Laidlaw et al. (2021) adopts neural perceptual threat model. Except for slightly behind Wang et al. (2023) in $l_\infty = 8/255$, ours outperforms all other methods significantly against multiple unseen attacks including the practical black-box Square Attack, although we only use standard training with clean images. Indeed, robustness against multiple attack models should be vital for applications since we can't assume the attack will follow the simulations conducted in the malicious sample generation in adversarial training methods. Unfortunately, most current works fail to generalize well to unseen attacks, even for specifically designed ones, Laidlaw et al. (2021) and Dai et al. (2022b). The results of $\sigma = 0$ in Table 1 highlight the importance of randomness in feature squeezing. Intuitively, more variants of inputs are very essential for robustness.It is also very interesting to note that there are very few drops in clean accuracy when we add noise, thanks to the great capacity of the deep neural network. Another significant advantage of ours is the computational cost shown in Table 2, where all other competitors in Table 2 are 1-5 orders of magnitude higher than ours.

As our algorithm is random in nature, we also adopt the EOT-test as shown in Table 3. There are some drops in accuracy, however, it is still much better than others for $l_1 = 12$ and $l_2 = 2$, which are very stable even under EOT-100. In fact, they get saturated at EOT-20, and due to some randomness,

Table 2: Computational cost comparison. Excluding the cost of gathering extra data, the training cost in #Cost is roughly the product of #Epochs(training epochs), #Extra, and #PGD(pgd steps adopted in adversarial inputs generation) with respect to ours, i.e., 50K inputs and 100 epochs of standard training, which is denoted by 1.

| Paper | #Extra | #Epochs | #PGD | #Cost |
|---|---|---|---|---|
| Wang et al. (2023) | 20M | 2400 | 10 | $9.6 \times 10^4$ |
| Gowal et al. (2021) | 100M | 2000 | 10 | $4 \times 10^5$ |
| Dai et al. (2022a) | 6M | 200 | 10 | $2.4 \times 10^3$ |
| Wang et al. (2023) | 50M | 1600 | 10 | $1.6 \times 10^5$ |
| Rebuffi et al. (2021) | 1M | 800 | 10 | $1.6 \times 10^3$ |
| Laidlaw et al. (2021) | 0 | 100 | 15 | 15 |
| Dai et al. (2022b) | 0 | 200 | 30 | 60 |
| Ours | 0 | 100 | 0 | 1 |

Table 3: The EOT accuracy of iterations at 1, 20 and 100 respectively for APGD-ce attack for 1000 images in CIFAR-10. EOT-20* stands for transfer attack.

| Attacks | EOT-1 | EOT-20 | EOT-100 | EOT-20* |
|---|---|---|---|---|
| $l_\infty$-$\epsilon$=8/255 | 69.40 | 58.20 | 56.90 | 55.60 |
| $l_\infty$-$\epsilon$=16/255 | 36.90 | 21.30 | 20.60 | 20.30 |
| $l_1$-$\epsilon$=12 | 76.70 | 55.70 | 56.20 | 77.10 |
| $l_2$-$\epsilon$=2 | 19.90 | 12.50 | 13.20 | 53.20 |

EOT-100 could be sometimes slightly better than EOT-20. The transfer attack is also conducted where we use the setting of the training to generate the EOT-20 adversarial examples. Even though we use the BPDA-like treatment in reciprocal and Sigmoid, the transfer attack is largely weaker than the EOT-20 operated on the net in the test itself. Note that EOT incurs a large computational cost, so actually, it is unfair to compare the robustness of networks without computation constraints

Finally, it should be noted that Jiang et al. (2023) proposed a dedicated adversarial training against $l_1$ attack, which achieves clean accuracy 76.14 and AA-$l_1$ 50.27, lower than ours 81.95 and 68.96. The only difference is that they use PreactResNet18, while ours use WideResNet-28-10, but presumably, this can not account for the large performance gap. This substantiates the super advantage of our approach when taking into account that our EOT-100 is around 55 and there is no direct relation with $l_1$ attack throughout our design.

### 5.1.2 ONE-PIXEL SHORTCUT

Although our approach is motivated for adversarial defense, it turns out ours is much less impacted by OPS without any data augmentation. Following the OPS Wu et al. (2023), we also choose ResNet-18 and all training settings are exactly the same as WideResNet-28-10 except for $\tau = 0.3$. Ours exceeds others by 40+. Please refer to Table 10 of Appendix for more results.

### 5.2 IMAGENET

ImageNet is the most challenging dataset for adversarial defense, and **there is no work dealing with unseen attacks even with adversarial training.** In this paper, ImageNet only refers to ImageNet-1k without explicit clarification, and robustness is only evaluated on the 5000 images of the ImageNet validation set as in RobustBench (Croce et al., 2021). For simplicity, we choose the architecture of ConvNeXt-T + ConvStem in Singh et al. (2023) with $\kappa = 6$, $\tau = 1.2$ and $\beta = 0.01$. It turns out that $S$ in Equation 8 quickly goes to nearly zero in one epoch. Our training scheme is very simple. All parameters are randomly initialized, followed by standard training for 50 epochs with heavy augmentations without CutMix (Yun et al., 2019) and MixUp (Zhang et al., 2018), as these will undermine the viability of our sampling matrix. While for the same ConvNeXt-T + ConvStem in Singh et al. (2023), although ConvStem is randomly initialized, the ConvNeXt-T part is from a strong pre-trained model which usually takes about 300 epochs. Thus the whole network needs extra

Table 4: AutoAttack comparison on ImagetNet. $l_\infty$-$\epsilon$=4/255, $l_1$-$\epsilon$=75 and $l_2$-$\epsilon$=2. AA-AVE stands for the average of AA. The bold indicates the best for each column. * denote models that are pre-trained with ImageNet-21k.For clean and AA, we report mean and standard deviation for five runs. **Reemphasize that there is no work for unseen attacks even with adversarial training, and all competitors in the table are based on adversarial training with $l_\infty$ attack.**

| Architecture | Clean | AA-$l_\infty$ | AA-$l_1$ | AA-$l_2$ | AA-AVE | Square-$l_\infty$ | Square-$l_1$, $l_2$ |
|---|---|---|---|---|---|---|---|
| **ConvNeXt-T + ConvStem** | | | | | | | |
| Singh et al. (2023) | 72.74 | 49.46 | 24.50 | 48.40 | 40.79 | 63.42 | 49.40 68.06 |
| Ours(mean) | 68.09 | 39.41 | **67.25** | **62.50** | **56.39** | 67.90 | **68.20** 67.40 |
| Ours(std) | 0.17 | 0.43 | 0.23 | 0.40 | 0.18 | | |
| Ours($\sigma = 0$) | 68.22 | 1.34 | 0.04 | 0.04 | 0.47 | | |
| **Swin-L** | | | | | | | |
| Liu et al. (2023b)* | 78.92 | **59.56** | 26.88 | 52.02 | 46.15 | **70.38** | 55.52 **74.18** |
| **ConvNeXt-L** | | | | | | | |
| Liu et al. (2023b)* | 78.02 | 58.48 | 26.18 | 52.22 | 45.63 | 70.12 | 54.40 72.86 |
| **ConvNeXt-L+ConvStem** | | | | | | | |
| Singh et al. (2023) | 77.00 | 57.70 | 22.38 | 47.02 | 42.37 | 69.66 | 54.18 72.80 |

standard training for 100 epochs to get good clean accuracy, followed by 300 epochs of adversarial training with 2-step APGD. So the total cost is up to $300 + 100 + 300 \times (2 \ (for\ APGD\ steps) + 1 \ (for\ weights\ update)) = 1300$, which is around $1300/50 = 26$ times bigger than ours.

As shown in Table 4, ours beats (Singh et al., 2023) by a large margin in almost all tests except $l_\infty$-$\epsilon$=4/255. To be more solid, we also compare with other methods of more sophisticated architectures, including Swin-L and ConvNeXt-L in Liu et al. (2023b). For AA, we are behind $l_\infty$-$\epsilon$=4/255, but much better in $l_1$-$\epsilon$=75 and $l_2$-$\epsilon$=2. On average, ours beats the second place Swin-L by 10, while for Square, only slightly behind on Square Attack $l_\infty$-$\epsilon$=4/255 and $l_2$-$\epsilon$=2. Again, a similar conclusion can also be drawn based on the results of $\sigma = 0$. Our specially designed input layer changes the input $x$ into $x'$ that are extremely squeezed. On one hand, it poses a great challenge to the network; while on the other hand, it improves the robustness. Some of the example feature maps in our input layers are listed in Figure 2 of Appendix.

Regarding EOT tests in Table 6, the negative impacts on robust accuracy are almost negligible for $l_1$, while for $l_\infty$ and $l_2$ there is a relatively high drop. As in CIFAR-10, we also conducted the transfer attack EOT-20* using the EOT-20 examples generated from the net in training. It seems that the performance is quite similar with the EOT-20 by itself, which is different from Table 3. This might due to the sophisticated ConvNeXt-T + ConvStem. We stress again here that in fact, every defense is weak given sufficient computational resources. As shown in Table 5, we increase a query limit of Square Attack from 5K used in AutoAttack to 50K denoted as Enhanced-Square, and there are up to 9% decrease in robust accuracy for Singh et al. (2023) and Liu et al. (2023b). Interestingly, because of randomness and the last-move strategy, ours stands up. Due to resource constraints, only 500 images are evaluated.

Table 5: Square Attack comparison on 500 images on validation set of ImagetNet instead of 5000 on Table 4. $l_\infty$-$\epsilon$=4/255, $l_1$-$\epsilon$=75 and $l_2$-$\epsilon$=2. The bold indicates the best for each column. * denote models that are pre-trained with ImageNet-21k. The iterations are 5K for Square Attack, and 50K for E-Square (Enhanced-Square Attack).

| Architecture | Clean | Square-$l_\infty$ | Square-$l_1$, $l_2$ | E-Square-$l_\infty$ | E-Square-$l_1$, $l_2$ |
|---|---|---|---|---|---|
| **ConvNeXt-T + ConvStem** | | | | | |
| Singh et al. (2023) | 72.20 | 65.80 | 51.20 69.00 | 61.60 | 42.00 66.80 |
| Ours | 69.80 | 69.40 | **68.60** 68.80 | **69.40** | **68.60** 69.00 |
| **Swin-L** | | | | | |
| Liu et al. (2023b)* | 79.80 | **70.60** | 55.00 **74.40** | 66.40 | 46.80 **71.80** |

Table 6: The EOT accuracy of iterations at 1, 20 and 100 respectively for APGD-ce attack for 500 images in ImageNet. EOT-20* stands for transfer attack.

| Attacks | EOT-1 | EOT-20 | EOT-100 | EOT-20* |
|---|---|---|---|---|
| $l_\infty$-$\epsilon$=4/255 | 42.80 | 32.40 | 29.20 | 33.20 |
| $l_1$-$\epsilon$=75 | 68.80 | 67.20 | 63.60 | 67.20 |
| $l_2$-$\epsilon$=2 | 65.60 | 54.00 | 50.80 | 53.00 |

Table 7: Performance of Clean for the whole test set and EOT-20 under APGD-ce attack for 1000 images in CIFAR-10 and 500 in ImageNet with different training strategies. T* denotes the training through the framework in test, while T for our method.

| Dataset | Attacks | T | T* |
|---|---|---|---|
| **CIFAR-10** | Clean | 82.01 | 81.65 |
| | $l_\infty$-$\epsilon$=8/255 | 58.20 | 50.10 |
| | $l_\infty$-$\epsilon$=16/255 | 21.30 | 16.00 |
| | $l_1$-$\epsilon$=12 | 55.70 | 38.20 |
| | $l_2$-$\epsilon$=2 | 12.50 | 10.10 |
| **ImageNet** | Clean | 68.09 | 67.86 |
| | $l_\infty$-$\epsilon$=4/255 | 32.40 | 34.00 |
| | $l_1$-$\epsilon$=75 | 67.20 | 68.60 |
| | $l_2$-$\epsilon$=2 | 54.00 | 54.40 |

It should be noted here that APGD-ce and APGD-dlr never give up searching for a better sample even though it already finds an effective one in the early iterations. For square, on the contrary, it will not search for a better sample once it already finds an effective one in the early iterations. Unfortunately, the example is very unstable and if fed once again, it will be classified correctly.

## 5.3 ABLATION STUDIES

As our approach is very special in that the test framework is different from the training one, a natural concern will arise: "Why not directly train in the configuration of the test?" In essence, this can be regarded as special case of our training framework where the parameters of all the CONV3×3 for sampling matrix $\hat{x}$ are manually set to be zero, then the reciprocal will become very big, and the Sigmoid is approximately equal to Sign. Accordingly there is no need to have $S$ in loss Equation 8. However, very importantly, we still need to have $\alpha$ in the first cross-entropy term to have a stable training, which shows exactly this is just the special case of our more general training setting. We adopt this training scheme in both CIFAR-10 and ImageNet, and make some comparisons of EOT-20 under APGD-ce attack with our method. The results are shown in Table 7. For CIFAR-10, it is very interesting to notice that training in the configuration of the test is always less robust than ours, especially for $l_1$-$\epsilon$=12, around 20 in performance gap, which indicates that our specially designed input layer can make the training very efficient. However, for ImagetNet, the performance is quite similar, this might be due to the small $\beta = 0.01$ in Equation 7 and quick convergence of $S$ to nearly zero in Equation 8.

## 6 THEORETICAL ANALYSIS

Although we have shown excellent experimental performance, one may still wonder why it is possible, especially for those uncomfortable with randomness. Here, we remove randomness and transform the test framework with random noise into a deterministic one. More specifically, we feed the $N$ copies of the same test image to the test framework, each with a different but fixed seed of noise, and then the average logits of $N$ outputs are used to get the final classification. It might be possible that our training scheme implements implicit adversarial training due to the added random noise; however, it is unclear how it relates to test robust accuracy. It appears that the feature squeezing is beneficial in this regard. Indeed, a deep connection exists between feature squeezing and Rademacher complexity for

Table 8: Accuracy for APGD-ce attack for the 1000 images from CIFAR-10 in both training and test sets (in bold) with different $N$.

| $N$ | Clean | $l_\infty$-8/255 | $l_\infty$-16/255 | $l_1$-12 | $l_2$-2 |
|-----|-------|------------------|-------------------|----------|---------|
| 5 | 100.00 **85.80** | 41.00 **32.30** | 6.70 **5.40** | 23.00 **18.40** | 1.20 **1.60** |
| 10 | 100.00 **85.20** | 45.30 **33.50** | 7.00 **5.40** | 26.40 **22.00** | 1.40 **1.60** |
| 15 | 100.00 **85.40** | 47.40 **34.20** | 7.70 **5.60** | 30.30 **23.70** | 1.20 **2.00** |
| 20 | 100.00 **85.50** | 48.30 **35.50** | 7.70 **5.60** | 30.60 **24.40** | 1.30 **2.10** |

Table 9: Accuracy for APGD-ce attack for the 500 images from ImageNet in both training and test sets (in bold) with different $N$.

| $N$ | Clean | $l_\infty$-4/255 | $l_1$-75 | $l_2$-2 |
|-----|-------|------------------|----------|---------|
| 5 | 82.60 **70.60** | 5.20 **4.20** | 26.60 **22.40** | 18.20 **13.00** |
| 10 | 83.20 **69.80** | 8.00 **6.40** | 40.00 **34.00** | 27.80 **21.40** |
| 15 | 83.60 **70.00** | 9.40 **8.00** | 46.20 **37.20** | 32.80 **26.20** |
| 20 | 83.80 **69.80** | 11.40 **9.60** | 46.80 **39.00** | 35.20 **28.80** |

adversarially robust generalization in Yin et al. (2019). Let $x \in X \subseteq R^d$ and $y \in Y \subseteq \{-1, 1\}$ be the feature and label spaces, and the training set $\{(x_1, y_1), (x_2, y_2), ..., (x_n, y_n)\}$. The experimental Rademacher complexity for the function class $F$ is defined as :

$$R_s(F) = \frac{1}{n} E_\sigma \left[ \sup_{f \in F} \sum_{i=1}^{n} \sigma_i f(x_i) \right]$$

where $\sigma_1, \sigma_2, ..., \sigma_n$ are independent random variables uniformly chosen from $\{-1, 1\}$, and $E_\sigma$ for expectation. The main result is as follows:

**Theorem 1.** *Let* $F := \left\{ \langle w, x \rangle : \|w\|_p \leq W \right\}$ *and* $\tilde{F} := \left\{ \min_{x' \in B_x^\infty(\varepsilon)} y \langle w, x' \rangle : \|w\|_p \leq W \right\}$. *Suppose that* $\frac{1}{p} + \frac{1}{q} = 1$. *Then there exists a universal constant* $c \in (0, 1)$ *such that*

$$\frac{c}{2} \left( R_s(F) + \varepsilon W \frac{d^{\frac{1}{q}}}{\sqrt{n}} \right) \leq R_s(\tilde{F}) \leq R_s(F) + \varepsilon W \frac{d^{\frac{1}{q}}}{\sqrt{n}}.$$

This indicates that experimental Rademacher complexity $R_s(\tilde{F})$ for the adversarial settings depends on the input dimension, which in turn will affect the generalization ability of the adversarial learning algorithm. We reduce the input dimension by transforming the pixels of the input image to the extreme values of 0 and 1, so test adversarial accuracy should not be far away from the training one, verified by our experiments with maxim discrepancy around 10 shown in Tables 8 and 9. Interestingly, as expected, non-trivial training robust accuracy is achieved through the implicit adversarial training with our specially designed input layer with the random noise, and according to Theorem 1, thanks to the feature squeezing, test robust accuracy also keeps up. Indeed, the overall performance is comparable with that of Dai et al. (2022b) and Laidlaw et al. (2021) for CIFAR-10, the only two approaches with similar motivations for unseen attacks but with training costs 60 and 15 times as of ours. For ImageNet, interestingly, ours achieves the highest $l_1$ accuracy compared with competitors in Table 4.

## 7  SUMMARY

In this paper, we proposed an efficient and effective method for unseen attacks only through standard training. To our knowledge, this is the only paper that falls within this category. Besides thorough experiments, we justify our excellence from the perspective of Rademacher complexity.

There are some limitations to this approach. Firstly, clean accuracy is not very good. Secondly, it is ineffective in defending against $l_\infty$ attack for ImageNet. Thirdly, there is no strong theoretical robustness guarantee.

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

## A  APPENDIX

### A.1  SOURCE CODES AND PRE-TRAINED MODELS

1. CIFAR-10
   `https://rfsq.obs.myhuaweicloud.com/CIFAR10.zip`
2. ImageNet
   `https://rfsq.obs.myhuaweicloud.com/imagenet.zip`

### A.2  LAST MOVE

The robust evaluation with the last move consists of two steps:

1. Adversarial image=Attack(cleanimage, model)
2. label =Model(Adversarial image)

In Step 1, we get the adversarial image based on the attack algorithm and clean image, and in Step 2, we evaluate our model on this adversarial image. Attack(clean image, model) may itself report some accuracy, which is not correct due to unstable adversarial images. For example, the attack can do nothing. It just sends the same clean image many times. Since our model is random, it may give a wrong prediction in one try. In other words, it may look like this. However, the attack cannot be

---

**Algorithm 1** Attack

---

**Input:** clean image, Model
**output:** adversarial image
 1: label=Model(clean image)
 2: **while** label is correct **do**
 3:     label=Model(clean image)
 4: **end while**
 5: print("successful attack")
 6: **return**  clean image

---

claimed to be successful, since once fed again, our model will predict correctly with high probability. So essentially, as stated in abstract in Däubener & Fischer (2022) "an adversarial attack is calculated based on one set of samples and applied to the prediction defined by another set of samples."

## A.3 OPS

Table 10: Performance under One-Pixel Shortcut on ResNet-18 for different training strategies. The first two rows are excerpted from Wu et al. (2023). $l_\infty$ AT stands for adversarial training with $l_\infty$=8/255.

| Training Strategy | Clean | OPS |
|---|---|---|
| Standard | 94.01 | 15.56 |
| $l_\infty$ AT | 82.72 | 11.08 |
| Ours | 82.58 | 56.44 |

## A.4 FEATURE MAP

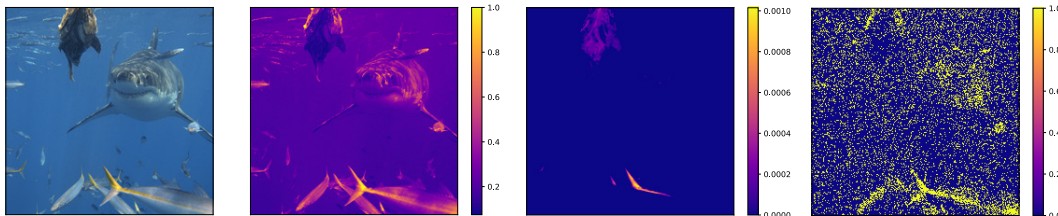

Figure 2: From the left to right are the great-white-shark $x$; the red channel; and the corresponding sampling matrix $\hat{x}$ and the final output $x'$ where the continuous patterns are highly squeezed into two extreme values, 0 and 1, due to very small $\hat{x}$. Blue and green channels share a similar situation.

