# OpenReview forum: "Randomized Feature Squeezing against  Unseen Attacks without Adversarial Training"
_ICLR.cc/2025/Conference — ICLR 2025 Conference Withdrawn Submission_

### Official Review · Reviewer_EZ39 · 2024-10-30

**Soundness:** 2
**Presentation:** 2
**Contribution:** 2
**Rating:** 5
**Confidence:** 4

**Summary:**

In this paper, the authors propose a novel approach that enables training robust networks using only standard training on clean images, without awareness of the attacker's strategy. They introduce a specially designed input layer that implements randomized feature squeezing to mitigate adversarial perturbations. This method demonstrates state-of-the-art robustness against unseen attacks while significantly reducing computational expenses. Experiments on CIFAR-10 and ImageNet validate its effectiveness, and it also defends against training data attacks, offering a potential solution for OPS attacks without data augmentation.

**Strengths:**

1.	The proposed approach allows for the training of robust networks solely using clean images without prior knowledge about the attacker's strategy. This makes it more practical and accessible for real-world applications.
2.	It achieves strong robustness against unseen attacks with lower computational costs and only requires 100/50 epochs of training on CIFAR-10 and ImageNet.

**Weaknesses:**

1.	The authors do not provide an explanation for why the proposed method (randomized feature squeezing) effectively defends against unseen attacks. Additionally, the designed loss function does not highlight aspects related to adversarial defense. The authors should theoretically or experimentally validate their defense principles from the perspective of features.
2.	The authors overlook comparisons with important defense methods such as image purification and denoising, including DiffPure [1] and NRP [2], which also do not need to train the classifier. They should include corresponding experiments.
[1] Nie W, Guo B, Huang Y, et al. Diffusion models for adversarial purification[J]. arXiv preprint arXiv:2205.07460, 2022.
[2] Naseer M, Khan S, Hayat M, et al. A self-supervised approach for adversarial robustness. In the proceedings of the IEEE/CVF Conference on Computer Vision and Pattern Recognition. (CVPR’20) 2020: 262-271.
3.	The authors do not explain how the several parameters of  the Input Layer mentioned in Section 4.1 and 4.2 are selected, nor how these parameters affect the method's performance. More ablation experiments should be included to address this issue.
4.	The authors lack detailed analysis and explanation of the experimental results of defend against OPS. Additional key insights into these findings are needed to strengthen the discussion.

**Questions:**

All the questions are included in Weaknesses.

---

> ### Author Response · Authors · 2024-11-23
>
> “Diffusion models for adversarial purification” is compared in lines 123-128.
>
> The robust evaluation of “A self-supervised approach for adversarial robustness” is wrong. Since the purify net is differentiable, it is unnecessary to approximate it with BPDA as JPEG preprocessing. *We download the pre-trained model and evaluate it against pgd correctly, and the robust accuracy is 0.*
>
>
> More ablation experiments.
>
> The theoretical analysis based on Rademacher complexity is much more important than more ablation experiments, and iclr only allows 10 pages.
>
> detailed analysis and explanation of the experimental results of defend against OPS.
>
> This is again due to the random featured squeezing. Since we transform all pixels to 1 or 0, the pixel chosen by the ops can not stand out from its neighbors.

---

> > ### Comment · Reviewer_EZ39 · 2024-11-26
> >
> > Thanks for the authors' reply. My concerns are basically solved, and  I will raise the score to 5. But I still think this work has shortcomings in the experimental setup and related discussion analysis.

---

> > > ### Author Response · Authors · 2024-11-26
> > >
> > > Thank you very much. We will try our best to enhance the paper as you suggest.

---

### Official Review · Reviewer_ihbk · 2024-11-02

**Soundness:** 3
**Presentation:** 2
**Contribution:** 3
**Rating:** 3
**Confidence:** 5

**Summary:**

In this paper, the authors propose a novel approach to train a robust network without the need for prior knowledge about attackers or adversarial training. It achieves this by adding a specially designed network input layer that performs randomized feature squeezing on clean images. The method shows state-of-the-art robustness against various unseen attacks in terms of computational cost in CIFAR-10 and ImageNet datasets.

**Strengths:**

1.	The topic of this paper, i.e., defending against adversarial examples solely via training with clean samples, is both meaningful and challenging.
2.	The designed input layer can be easily integrated into different networks like WideResNet and ConvNeXt to boost their performance, demonstrating its potential for wide application across various network architectures.
3.	This paper is easy to follow.

**Weaknesses:**

1.	Even though the authors claim that the proposed method can effectively defend against adversarial samples based on the $l_{p}$ norm constraint, they have not demonstrated its performance against another mainstream type of generated adversarial examples, such as those from GAN-based attack methods [1, 2] and diffusion model-based attack methods [3, 4]. The authors are required to provide the corresponding defense results to prove the effectiveness of the proposed method.
2.	In the method section, the authors only provided the implementation steps for adding the input layer, without any analysis of the effectiveness of this method, such as theoretical derivations and experimental analyses.
3.	The experimental setup in this paper lacks persuasiveness. The attacks defended in this paper,  FAB-attack and Square Attack, are rather outdated. The authors need to supplement the experimental results of the most recently published adversarial attacks based on the norm constraint in top conferences within the past two years.
4.	As mentioned in the SUMMARY section, the defense method proposed in this paper does not even guarantee its effectiveness under the norm constraint. The overall performance, including both accuracy and robustness, is somewhat lacking. I find the claim that the proposed method can defend against samples merely through standard training with clean samples unconvincing. The commonly recognized understanding in the current community is that adversarial examples originate from the inherent vulnerability of deep learning models. Given that the authors propose to defend against adversarial samples under the settings of this paper, they should prove it through theoretical analysis on robustness rather than simply presenting some experimental results.


Reference

[1] Generative Adversarial Perturbations. CVPR 2018.

[2] Downstream-agnostic Adversarial Examples. ICCV 2023.

[3] AdvDiffuser: Natural Adversarial Example Synthesis with Diffusion Models. ICCV 2023.

[4] Diffusion Models for Imperceptible and Transferable Adversarial Attack. TPAMI 2024.

**Questions:**

See Weakness

---

> ### Author Response · Authors · 2024-11-23
>
> Please see the general response.

---

> > ### Comment · Reviewer_ihbk · 2024-11-24
> > **Responses by Reviewer ihbk**
> >
> > Thanks for responses. After a meticulous review of all the reviewers' comments and the rebuttal provided by the author, it is evident that the proposed method in this paper suffers from a lack of effectiveness and applicability, and the experimental evaluation is conspicuously inadequate. Consequently, I believe this work demands further refinement to meet the standards for publication.

---

### Official Review · Reviewer_Tjmg · 2024-11-02

**Soundness:** 3
**Presentation:** 3
**Contribution:** 3
**Rating:** 8
**Confidence:** 4

**Summary:**

This paper proposes a new robust training method that does not require adversarial training. The proposed method utilizes a special input layer that processes the input in two ways: the first way corrupts the input with dependent Gaussian noise, and the second one convolves the input with a 3x3 2d kernel and applies ReLU, then the values of the resulting convolution are inverted. The results of two ways are multiplied, and sigmoid activation is applied to the resulting multiplication. Then, the output of the input layer is forwarded to a standard image classification network. The loss function is modified in a way that convolution output ($\hat{x}$) is small. Therefore, it has an additional term besides weighted standard cross entropy loss. During inference, the second way in the input layer, convolution is dropped and sigmoid activation is replaced with sign function. Experiments are carried out on CIFAR-10 and ImageNet datasets with AutoAttack (AA) on different norms. The results show that clean accuracy slightly dropped and robust accuracy on AA $\ell_\infty$ is slightly behind the adversarially trained (AT) models. However, the proposed method outperforms AT models on other norms $\ell_\infty$, $\ell_1$ and $\ell_2$. In addition, the paper also shows that this proposed robust training has less impact on unlearnable examples (one-pixel shortcut).

**Strengths:**

- The paper provides a new perspective on randomization in the adversarial defense, although randomization is notoriously known to be vulnerable in defense research.
- The paper justifies the design choices with intuition that gives good insights.
- The paper highlights the last move strategy not to have wrong robust accuracy.
- The paper also shows unlearnable examples less impact the proposed training method.
- The paper clearly presents the limitations of the proposed training.
- Overall, this new way of robust training is interesting and brings many educational values.

**Weaknesses:**

- The second contribution says the proposed method is the only work that does not require prior knowledge about the attacks with standard training with clean images. That is not completely true. Please refer to BaRT[1] and LINAC [2].
BaRT uses a set of random transforms with random parameters and standard training, it does not require prior knowledge about the attacks. LINAC uses implicit neural representation with secret key and standard training (no prior knowledge about the attacks). The proposed method may be more related to LINAC.
- The paper does not explicitly define a threat model, unlike adversarial training. It seems the proposed method is robust against AA under $\ell_\infty$, $\ell_1$ and $\ell_2$. I am a bit skeptical of the robustness. The reason is that there must be some bound that the Gaussian noise simulation in the training can cover. This is not clearly known from the current experiments. It would be better to know what type of noise and how much noise the proposed method is robust against.
- The paper does not compare with other attack agnostic defenses such as [1] and [2].
- The paper does not consider any adaptive attack apart from EoT, which is important for a defense evaluation. Please consider parametric bypass approximation (PBA) and discuss potential adaptive adversaries.
- Experiments are limited to AA attacks. Randomized defenses should especially be intensively evaluated with more black-box attacks. Please consider SPSA, N-Attack, and one pixel attack.

[1] https://openaccess.thecvf.com/content_CVPR_2019/papers/Raff_Barrage_of_Random_Transforms_for_Adversarially_Robust_Defense_CVPR_2019_paper.pdf
[2] https://proceedings.mlr.press/v162/rusu22a.html

**Questions:**

I am still skeptical of the robustness. If you provide more evidence of robustness, I will increase the score.
- In the AA framework, there are two versions: standard and random. What version did you use for evaluation?
- Experiment results are shown on noise budget 8/255, 16/255 for CIFAR-10, and 4/255 for ImageNet. What happens if you increase the noise budget? Is there any relation between noise simulation in the training and attack noise budget?
- Square attack alone for the black box is not enough. What about at least SPSA [3] and N-Attack [4].
- Will the proposed method be robust against attack methods with masked perturbation? For example, you can apply PGD only on certain region and mask the other out. I am curious about the proposed noise simulation generalizability.
- It is important to consider adaptive attacks. At the very least, parametric bypass approximation (PBA) from LINAC [2] should be considered.
- From the results, the robustness of the proposed training is comparable to adversarial training. Will the proposed robust model have generative gradients as in adversarially trained models? It would be better if you could provide some analysis/visualization of the proposed robust model.

[3] https://proceedings.mlr.press/v80/uesato18a.html
[4] https://proceedings.mlr.press/v97/li19g/li19g.pdf

Minor Comment
In Section 5.1.1, the end of second last paragraph is missing the full stop.

---

> ### Author Response · Authors · 2024-11-23
>
> BaRT[1] adopts random transformations that pose a high challenge to robust evaluation. Indeed “Demystifying the Adversarial Robustness of Random Transformation Defenses, icml2022” gets a much worse robust accuracy than BaRT[1]. *Moreover,  we download the pre-trained model and transform the BaRT to the deterministic one as we did in our paper, the robust accuracy is 0 against the pgd attack.*
>
> LINAC [2] is very different from ours in that there is a secure key that is not open to the attacker, so the evaluation with a complete white box attack is not conducted in LINAC [2]. *We tested the model with the same key as the defender, with only 15% robust accuracy against pgd-10. In our case, the attack has full access to the defender's source code.*
>
> The Gaussian noise in the training has two effects. On the one hand, it simulates the adversarial attack to some extent; but on the other, it destroys the clean accuracy, so the sigma should not be high. Of course, it cannot implement the exact adversarial attack, so there is still a gap between robust accuracy and clean one, and the gap will become large for stronger attacks with a larger budget.
>
> Parametric bypass approximation (PBA) applies only to key-based defense and has nothing to do with ours.
>
> Will the proposed method be robust against attack methods with masked perturbation?
>
> Indeed, the pgd attack with $l_1$ constraint already implements this. For imagenet, only 1.53% of total pixels will be affected, while  16.89%  for cifar-10.
>
> In the AA framework, there are two versions: standard and random. What version did you use for evaluation?
>
> AA for random contains apgd-ce and apgd-t with eot-20. For tables explicit with EOT, we use apgd-ce , since apgd-t has similar performance; Otherwise, we use the standard version.
>
> Will the proposed robust model have generative gradients as in adversarially trained models?
>
> No, it is just standard training with clean images. Please see the general response.
>
> | DataSet      | Clean/SPSA      |Clean/Nattack| Clean/One-pixel    |
> | ------------- | ------------- |------------- | ------------- |
> | Cifar-10 | 81/68 | 80/70 | 82/76 |
> | ImageNet | 72/70 | 72/66 | 75/75|
>
> Since we run tests independently, we get different clean accuracy.
>
> Minor Comment In Section 5.1.1, the end of second last paragraph is missing the full stop.
>
> Thank you very much.

---

> > ### Comment · Reviewer_Tjmg · 2024-11-24
> >
> > Thank you for the response and additional experiments on the blackbox attacks.
> > I carried out a quick experiment and the proposed defense is not robust against even a simple BPDA, so I am guessing the robustness claimed is from obfuscated gradients. Here is the attack that defeats the proposed defense.
> >
> > # BPDA
> > The main part that is responsible for robustness in the the proposed defense is Gaussian noise addition in the inference. To quickly vibe check, just change the scalar value 6 in `rand_x1` to 2. You can easily attack the proposed defense with PGD. To implement BPDA, replace `rand_x1` with identity function in your backward.
> >
> > >Parametric bypass approximation (PBA) applies only to key-based defense and has nothing to do with ours.
> >
> > Why PBA is nothing to do with the proposed defense since there is a non-differentiable sign function, how do I know the approximation used is better than neural network approximation? Maybe you have theoretical proof that you do not need PBA.
> >
> > # L1 attack
> > >Indeed, the pgd attack with constraint already implements this. For imagenet, only 1.53% of total pixels will be affected, while 16.89% for cifar-10.
> >
> > Well, I tested the SparseL1DecentAttack with 5% of noise addition. Here is the exact parameters.
> > ```
> > adversary = SparseL1DescentAttack(
> >     model, eps=1000., eps_iter=2*1000./40, nb_iter=100,
> >     rand_init=True, targeted=False)
> > ```
> > It worked as well. The noise is not visible to my eyes. Your threat model is not clear. Maybe you may say the noise budget is too high.

---

> > > ### Author Response · Authors · 2024-11-24
> > >
> > > Thank you very much for your prompt reply.
> > >
> > > To quickly vibe check, just change the scalar value 6 in rand_x1 to 2. You can easily attack the proposed defense with PGD. To implement BPDA, replace rand_x1 with identity function in your backward.
> > >
> > > You change the defense model. The model is pre-trained with 6, however, you change it to 2. Of course, the defense model can not be changed since it has already been deployed. let's say you change it to rand_x1=10000*torch.randn_like(x1).to(device), then clean accuracy will be 0 and there is no need to attack at all.
> > >
> > > This is just for quick response to this matter, and we will respond to other comments later.

---

> > > > ### Comment · Reviewer_Tjmg · 2024-11-24
> > > >
> > > > Thank you for the quick response.
> > > > I think it is better to follow the paper "On Evaluating Adversarial Robustness" (https://arxiv.org/abs/1902.06705). To convince the reviewers, you also need to discuss the adaptive attacks.
> > > > The point of changing the defense model **is not an attack**. It just gives me a clue to implement BPDA.
> > > > When I change rand_x1 to 2, the clean images are still classified correctly. I am not sure the threat model you consider cannot analyze the model by changing its parameter.

---

> > > > > ### Author Response · Authors · 2024-11-24
> > > > >
> > > > > I am not sure the threat model you consider cannot analyze the model by changing its parameter.
> > > > >
> > > > > Of course, it is possible to change its parameters. It is called a transfer attack since essentially the attack model is different from the defense model, which we have done in the paper although under different situations.
> > > > >
> > > > > But generally, since we know everything about the defense model, why we should bother to change its parameters and issue a transfer attack?
> > > > >
> > > > > Why PBA is nothing to do with the proposed defense since there is a non-differentiable sign function, how do I know the approximation used is better than neural network approximation
> > > > >
> > > > > The classifier adopted by key-based approach is ${f_\theta }(t(x))$, which
> > > > > operates on transformed inputs (i.e. on t(x) rather than on x), using a private key that is not accessible to the attacker. That is why we should take a PBA attack to approximate the t(X). In our case, everything is open, and why should we resort to PBA?
> > > > >
> > > > > adversary = SparseL1DescentAttack(
> > > > >     model, eps=1000., eps_iter=2*1000./40, nb_iter=100,
> > > > >     rand_init=True, targeted=False)
> > > > >
> > > > > We tested this on the 100 samples and the robust accuracy is 6%, however for Singh et al. (2023), the first row of Table 4, it is 4%. So ours is better than the competitors.

---

> > > > > > ### Comment · Reviewer_Tjmg · 2024-11-24
> > > > > >
> > > > > > Thank you for the quick response.
> > > > > >
> > > > > > >Of course, it is possible to change its parameters. It is called a transfer attack since essentially the attack model is different from the defense model, which we have done in the paper although under different situations.
> > > > > >
> > > > > > Sorry, I think there is a misunderstanding. I did not perform the transfer attack. I changed the parameter to see how the model behaves. Then, I just replaced `rand_x1` with an identity function. In my simple BPDA, your `rand_x1` value 6 is not changed. So, there is no transfer attack. The same model, the same parameters.
> > > > > >
> > > > > > >The classifier adopted by key-based approach is
> > > > > > , which operates on transformed inputs (i.e. on t(x) rather than on x), using a private key that is not accessible to the attacker. That is why we should take a PBA attack to approximate the t(X). In our case, everything is open, and why should we resort to PBA?
> > > > > >
> > > > > > I understand your point. But you have gradient approximation in the sign function and noise addition. If you consider the training data is available, my point is to learn a function (of your sign function with noise addition) like PBA. I do not have evidence that it will work. Wouldn't it be better to find possible adaptive attacks rather than ignoring it completely? Or maybe you are pretty sure that it will not work. I think you are taking an adaptive attack literally.
> > > > > >
> > > > > > >We tested this on the 100 samples and the robust accuracy is 6%, however for Singh et al. (2023), the first row of Table 4, it is 4%. So ours is better than the competitors.
> > > > > >
> > > > > > Thank you for the quick test. So, the proposed defense is 2% better than the adversarial training.
> > > > > >
> > > > > > **Please confirm the simple BPDA attack whether it breaks the proposed defense.**

---

> > > > ### Comment · Reviewer_hFEn · 2024-11-24
> > > >
> > > > Reviewer hFEn here. Successful attacks that bypass gradient obfuscation would typically find novel ways to modify the model to generate adversarial examples, and test them on the original defense model. For example, beyond what has been suggested by Reviewer Tjmg, could adversarial examples be generated, say with values ranging from 0.001 to 1000, and tested against your defending model?
> > > >
> > > > Finding practical and sensible ways to challenge your own defense beyond what is done in current attacks, particularly by identifying potential bypasses, will go a long way towards making your contribution more rigorous and useful.

---

> ### Author Response · Authors · 2024-11-24
>
> Thank you very much for your prompt reply over the weekend.
>
> --------------------------------------------------------------------------------------------------------------
> For PBA, we understand that you suggest building a differentiable proxy model and launching a transfer attack. However, this is out of the scope of this paper since there is no guide for the network structure of this model.
>
> ------------------------------------------------------------------------------------------
> In fact,  we did the initial assessment using the identity BPDA which is very common in practice. Please see the following tables for APGD-ce on 100 test samples. It will give obfuscated gradients on $l_2$, so we decided to use Sigmoid which also happens to be in
>  “One Man’s Trash is Another Man’s Treasure: Resisting Adversarial Examples by Adversarial Examples”, cvpr 2020
> .
>
> | CIFAR-10| $l_\infty$-8/255 | $l_1$-12 | $l_2$-2|
> |-------|-------|-------|-------|
> | Identity  | 68 |75 |40|
> | Sigmoid | 71 | 74 |22|
>
>
> | ImageNet| $l_\infty$-4/255 | $l_1$-75 | $l_2$-2|
> |-------|-------|-------|-------|
> | Identity  | 48 |72 |68|
> | Sigmoid | 46 | 73 |67|
>
> adversary = SparseL1DescentAttack( model, eps=1000., eps_iter=2*1000./40, nb_iter=100, rand_init=True, targeted=False)
>
> For this attack, the robust accuracy is 2% if identity is adopted. So maybe different BPDA should be implemented for the different attacks, however, the adversarial training approaches only achieve 4%, at the great cost of training. So it shows the great potential of ours.

---

> > ### Comment · Reviewer_Tjmg · 2024-11-25
> >
> > Dear authors,
> >
> > Thank you for the effort and clarification. I acknowledge the good work.
> > As I am trying to identify the capability of the proposed defense, I ran the test for 1000 images from NIPS 2017 adversarial example competition. The attack used is PGD with 40 steps and noise budget of 4/255. I run several times to avoid last move strategy.
> >
> > | Condition | Accuracy (%)|
> > | --- | --- |
> > | Clean | 79.9 |
> > | PGD | 35.6 |
> > | BPDA (identity) | 30.4 |
> > | BPDA (sigmoid) | 29.9 |
> >
> > I think your experiments also show a similar trend. How do we know there is no obfuscated gradients? In my opinion, I think it is important so that we can avoid a false sense of security. The accuracy may further drop slightly if I do PGD 100 steps. Due to the time constraint, I stick with PGD 40 steps.
> >
> > Since the proposed defense is designed differently, it is not helpful to follow the attacks from the adversarial training. It seems that the proposed defense is effective to remove high frequency adversarial noise. To test my hypothesis, I ran a simple test of 10 images from ImageNet validation set (all 10 images are classified correctly by the proposed defense). I used 4/255 noise budget with only 20 PGD steps and implement low frequency perturbation [a]. 8/10 adversarial examples successfully fooled the proposed defense, the 2 images that failed have only single object with plain background, therefore, low frequency perturbation was not successful. However, I believe if I perturbed carefully, the 2 images may also fool the defense.
> >
> > Although I appreciate the proposed defense and acknowledge the great work, the proposed defense in its current form is not clear of its capability. Therefore, I still cannot increase the score.
> >
> > [a] https://proceedings.mlr.press/v115/guo20a/guo20a.pdf

---

> ### Author Response · Authors · 2024-11-26
>
> Thank you very much for your engagement in the discussion and the experiments you have run.
>
> I think your experiments also show a similar trend. How do we know there is no obfuscated gradients?
>
> So, we use exactly BPDA (sigmoid) in this paper. Is there anything wrong?
>
> To test my hypothesis, I ran a simple test of 10 images from ImageNet validation set (all 10 images are classified correctly by the proposed defense). ….
>
> Thanks for your experiments, which help identify the pitfalls of our defense. Every defense has its pros and cons. Moreover, as discovered in “INEQUALITY PHENOMENON IN l-inf ADVERSARIAL TRAINING, AND ITS UNREALIZED THREATS,(iclr 2023)” the standard trained model can be more reliable than adversarial ones against occlusion attacks. Our key contribution is improving the robustness only with standard training with clean images, at least for black-box attacks.

---

> > ### Comment · Reviewer_Tjmg · 2024-11-26
> >
> > Dear authors,
> >
> > Thank you for the continuous engagement.
> >
> > >So, we use exactly BPDA (sigmoid) in this paper. Is there anything wrong?
> >
> > If my memory serves, you have mentioned only once of BPDA experiment in the paper (in the experiment section) saying EOT is better than the transfer attack for BPDA of reciprocal and sigmoid. Here is the exact sentence, "Even though we use the BPDA-like treatment in reciprocal and Sigmoid, the transfer attack is largely weaker than the EOT-20 operated on the net in the test itself.". As a reviewer, I have to verify that the proposed defense is actually robust. As you know, there are a lot of previous defenses broken due to obfuscated gradients. Even if you think your BPDA (sigmoid) is the best, I do not know there is still obfuscated gradients. To confirm with a known better approximation like (PBA), you think it is out of scope. Due to the time constraint, I could not verify it myself. So, nothing is wrong. I just do not know the proposed defense has more obfuscated gradients.
> >
> > >Thanks for your experiments, which help identify the pitfalls of our defense. Every defense has its pros and cons. Moreover, as discovered in “INEQUALITY PHENOMENON IN l-inf ADVERSARIAL TRAINING, AND ITS UNREALIZED THREATS,(iclr 2023)” the standard trained model can be more reliable than adversarial ones against occlusion attacks. Our key contribution is improving the robustness only with standard training with clean images, at least for black-box attacks.
> >
> > I agree. The threat model in the adversarial training is well defined. However, the robustness of the proposed defense is not clear.

---

> ### Author Response · Authors · 2024-11-26
>
> Thank you very much for your response.
>
> Here is the exact sentence, "Even though we use the BPDA-like treatment in reciprocal and Sigmoid, the transfer attack is largely weaker than the EOT-20 operated on the net in the test itself.".
>
> There is a big misunderstanding. We adopt all the tests in the paper with BPDA-sigmoid, which is clearly stated in lines 240-242, and implemented in our model code you run. This sentence means since our training and testing frameworks are different, people may wonder if it is possible to do the transfer attack. In the training framework, we also adopt BPDA to generate the adversarial samples for transfer attacks.
>
> in convnext_sampling.py
>
>         if not self.for_attack:
>             x2 = (x1 + rand_x1) * self.my_pow(sampling_weight)
>             x = self.my_sigmoid(x2)
>         else:
>             x = my_sign_class.apply(x1 + rand_x1)
>
>  the line: x = my_sign_class.apply(x1 + rand_x1) is just for attack
>
>
> class my_sign_class(torch.autograd.Function):
>
>     @staticmethod
>     def forward(ctx, input):
>         ctx.input = input
>         return (torch.sign(input) + 1) / 2
>
>     @staticmethod
>     def backward(ctx, grad_output):
>         scalar = 5.
>         input = ctx.input
>         temp = torch.sigmoid(scalar * input)
>         # return grad_output*1
>         return grad_output * temp * (1 - temp) * scalar
>
> This section exactly implements the BPDA-sigmoid for sign

---

> > ### Comment · Reviewer_Tjmg · 2024-11-26
> >
> > Thank you for the quick response.
> >
> > >There is a big misunderstanding. We adopt all the tests in the paper with BPDA-sigmoid, which is clearly stated in lines 240-242, and implemented in our model code you run. This sentence means since our training and testing frameworks are different, people may wonder if it is possible to do the transfer attack. In the training framework, we also adopt BPDA to generate the adversarial samples for transfer attacks.
> >
> > Thank you for the clarification. I did not mean you don't care about BPDA. I understood the explanation in line 240-242 and saw the `backward` implementation. My BPDA test did not use that code. I dissect your code. I separate noise addition and sign function as two functions, tested each one obfuscating ability. I was basically looking for which part is responsible for robustness and why my attack optimization failed. I keep asking the BPDA we tried are good enough or not. If we do not do a good job, we will not get a good evaluation.

---

> > > ### Comment · Reviewer_Tjmg · 2024-11-26
> > >
> > > Dear authors,
> > >
> > > After careful consideration, I increased the score. If the paper is accepted, you need to clearly explain what the proposed defense can and cannot do. I believe the proposed defense brings a new perspective to the adversarial machine learning research.

---

> > > > ### Author Response · Authors · 2024-11-26
> > > >
> > > > Thank you very much for your kind encouragement and help during the discussion. Of course, we will follow your instructions regardless of the paper's acceptance.

---

> > > > > ### Comment · Reviewer_Tjmg · 2024-11-26
> > > > >
> > > > > In the previous discussion, I tried to explain my thought process why I was skeptical. In case, if you want to know what I did in code.
> > > > >
> > > > > ```
> > > > > class Noise(nn.Module):
> > > > >     def __init__(self):
> > > > >         super().__init__()
> > > > >     def forward(self, x):
> > > > >         torch.seed()
> > > > >         rand_x1 = torch.clamp_max(6 * torch.abs(x), 1.2) * torch.randn_like(x).to(device)
> > > > >         x = x + rand_x1
> > > > >         return x
> > > > >
> > > > > class Sign(nn.Module):
> > > > >     def __init__(self):
> > > > >         super().__init__()
> > > > >     def forward(self, x):
> > > > >         x = (torch.sign(x) + 1) / 2
> > > > >         return x
> > > > >
> > > > > noise = Noise()
> > > > > sign = Sign()
> > > > >
> > > > > from advertorch.bpda import BPDAWrapper
> > > > > noise_bpda = BPDAWrapper(noise, forwardsub=lambda x: x)
> > > > > sign_bpda = BPDAWrapper(sign, forwardsub=lambda x: x)
> > > > > defended_model = nn.Sequential(noise_bpda, sign_bpda, model)
> > > > > ```
> > > > > On 1000 test images, the default model with your BPDA without any change gave me 35.6%, I removed the input processing layer outside and replace it with identity on backward, then it gave me 30.4%. Then, I know my BPDA is not good enough. Anyway, research is progressive and I should support. Thank you for the discussion.

---

> > > > > > ### Comment · Reviewer_Tjmg · 2024-11-26
> > > > > >
> > > > > > I used this code, to test low frequency perturbation.
> > > > > > ```
> > > > > > https://github.com/cg563/low-frequency-adversarial
> > > > > > ```

---

> ### Author Response · Authors · 2024-11-27
>
> Dear Reviewer Tjmg:
> ﻿
>
> When we practiced your code, we surprisingly found that plain PGD is stronger than AA APGD-ce, which we take for granted is much stronger. We have not tried plain PGD before.  It will make the robust evaluation incorrect.  The separation of the noise and sign is also very beneficial.
> ﻿
>
> We have decided to withdraw this paper. We appreciate your professionalism and support during the whole review process.
> ﻿
>
> Thanks
>
> Best regards

---

### Official Review · Reviewer_hFEn · 2024-11-04

**Soundness:** 1
**Presentation:** 1
**Contribution:** 2
**Rating:** 3
**Confidence:** 4

**Summary:**

This paper proposes to inject noise into the input image with randomized Gaussian noise and learns to perturb the image to near binarized values in order to promote robustness against test-time adversarial attacks.

**Strengths:**

The results appear to be good. It is also interesting to see papers that work on adversarial attacks with unseen threat models. However there are major concerns regarding its trained model robustness.

**Weaknesses:**

The results of the proposed method appear to be largely influenced by obfuscated gradients. This paper cites this issue in Line 224, but only for the training phase. The paper fails to recognize that Athalye et al.'s key contribution is they highlighted that the models themselves may not be robust because of gradient obfuscation. There are no result in this paper to lessen my worries regarding this concern.

Regarding unseen threat models, recent works typically employ other attacks that go beyond the $\ell_p$ boundaries. For instance, JPEG corruption [a], ReColorAdv [b], LPA [c], StAdv [d], FSA [e] and even methods that generate realistic natural adversarial examples [f].

[a]: Kang et al., Testing robustness against unforeseen adversaries. https://arxiv.org/abs/1908.08016
[b]: Laidlaw et al., Functional Adversarial Attacks. https://arxiv.org/abs/1906.00001
[c]: Laidlaw et al., Perceptual adversarial robustness: Defense against unseen threat models. ICLR, 2021.
[d]: Xiao et al., Spatially Transformed Adversarial Examples. ICLR 2018.
[e]: Xu et al., Towards feature space adversarial attack by style perturbation. AAAI 2021.
[f]: Chen et al., AdvDiffuser: Natural Adversarial Example Synthesis with Diffusion Models. ICCV 2023.

**Questions:**

- Could you address the major concern regarding gradient obfuscation?
- Could you test your model on existing attacks beyond the $\ell_p$ confines?

---

> ### Author Response · Authors · 2024-11-23
>
> Of course, we did BPDA in our test.
>
>
> As stated in section 4.1 of “Obfuscated Gradients Give a False Sense of Security:
> Circumventing Defenses to Adversarial Examples”, “To attack defenses where gradients are not readily available, we introduce a technique we call Backward Pass Differentiable Approximation (BPDA) ”
>
> As clearly stated in lines 240-242, since Sign is the only non-differentiable component in our test, the derivative of smooth Sigmoid(ax) is selected to implement Backward Pass Differentiable Approximation (BPDA), where a is chosen to get the worst robust accuracy to make sure our evaluation is legitimate. In this paper, a = 5. A similar treatment also appears in “One Man’s Trash is Another Man’s Treasure: Resisting Adversarial Examples by Adversarial Examples”, cvpr 2020. But we are more rigorous in that we test different a, and choose the one with the worst robust accuracy.

---

> > ### Comment · Reviewer_hFEn · 2024-11-24
> >
> > Thank you for your comment, however the question regarding unseen threat models remain unaddressed, and the paper would benefit from additional empirical results to show its effect, and analyses of the results to also gain a better understanding of how the proposed method works.

---

### Author Response · Authors · 2024-11-23
**General Response**

Thanks for the reviews.

Although not elaborated on in the title, we evaluate our defense against  ${l_p}$-attacks in this paper. So we may change the title to “Randomized Feature Squeezing against Unseen ${l_p}$-Attacks without Adversarial Training.” However, we believe this is interesting enough to deserve in-depth research, and in fact, the most sophisticated adversarial training approaches still perform poorly on AA.

Regarding theoretical guarantee/ unconvincing/ skeptical of the robustness / why the proposed method (randomized feature squeezing) effectively defends against unseen attacks;



“The commonly recognized understanding in the current community is that adversarial examples originate from the inherent vulnerability of deep learning models.”.
Yes. It is true. However, it doesn’t necessarily mean we cannot improve the robust accuracy only with standard training with clean images.


The key to success is the random feature squeezing. Suppose the input image dimension is M by N, and each pixel has 8 bits per channel. The total feature space is ${\left( {M \times N} \right)^{256 \times 256  \times 256}}$, while in our case,  only ${\left( {M \times N} \right)^{2\times 2\times 2}}$ at the cost of reduced clean accuracy. The introduced random noise will help explore the potential adversarial feature space. Its effectiveness is shown in the adversarial training accuracy in Tables 8 and 9. Moreover, the theory of experimental Rademacher complexity for the adversarial settings guarantees that the test adversarial accuracy should not be far away from the training one due to our extreme feature squeezing, so we also have good test robust accuracy.

All the experiments are conducted on only 100 samples for quick assessment in this rebuttal.

---

### Author Response · Authors · 2024-11-24

Dear AC

This paper is based only on standard training with clean images, which can perform similarly to or better than adversarial training.
﻿
If this paper gets rejected because it doesn't include experiments beyond AA, what about others that adopt expensive adversarial training and do not compare with the state-of-the-art in robust bench for every setting of l-p attacks such as ours?

Moreover, our approach is the only one that can deal with OPS, which has nothing to do with any suspicion of robust evaluation.

Thanks.

Best regards.

---

### Note · Authors · 2024-11-27

I have read and agree with the venue's withdrawal policy on behalf of myself and my co-authors.